# Insights into Atresia Ani Type IV in *Felis catus*: Preliminary Epidemiolocal Findings Associated with Surgery

**DOI:** 10.3390/ani14121738

**Published:** 2024-06-08

**Authors:** Diogo Nascimento, Pedro Azevedo, L. Miguel Carreira

**Affiliations:** 1Anjos of Assis Veterinary Medicine Centre—CMVAA, Rua D.ª Francisca da Azambuja Nº9 -9A, 2830-077 Barreiro, Portugal; diogofmnascimento98@gmail.com (D.N.); pedro.almeida.azevedo@gmail.com (P.A.); 2Faculty of American Laser Study Club—ALSC, Altamonte Springs, FL 32714, USA; 3Faculty of Veterinary Medicine, University of Lisbon (FMV/ULisboa), Av. da Universidade Técnica, 1300-477 Lisboa, Portugal; 4Interdisciplinary Centre for Research in Animal Health (CIISA), University of Lisbon, (FMV/ULisboa) Av. da Universidade Técnica, 1300-477 Lisboa, Portugal; 5Associate Laboratory for Animal and Veterinary Sciences (AL4AnimalS), 1300-477 Lisbon, Portugal

**Keywords:** cat, Atresia Ani Type IV, surgery, recto-vaginal fistula, epidemiology, insights, prognosis, post-surgery period

## Abstract

**Simple Summary:**

Atresia Ani (AA) is a rare congenital anomaly affecting rectal and anal development in companion animals, with its occurrence in cats being poorly documented. This study investigated Type IV Atresia Ani (which includes a recto-vaginal fistula) in female cats, focusing on surgical intervention as the primary treatment. Over 2 years, nine female cats with Type IV Atresia Ani were analyzed for age, body condition, and fistula length. Statistical analysis revealed that a body condition score of 3, an age range of 3 to 4 weeks, and a fistula length of 1 to 2 mm influenced fecal incontinence and anal stenosis development during the perioperative period. Incidence of this condition was 4.7% among the 192 cats evaluated. Cats with a body condition score of 3, an age of 3 to 4 weeks, and a fistula size of 1 to 2 mm showed better surgical outcomes. This study is the first to report the disease incidence in cats undergoing surgery, according to the authors’ knowledge.

**Abstract:**

Atresia Ani (AA) is a rare congenital anomaly in companion animals affecting rectal and anal development. Its incidence in cats remains unreported. This retrospective study aimed to characterize age, body condition, fistula size, and perioperative outcomes in Type IV Atresia Ani (with recto-vaginal fistula) in female cats. Surgical intervention is the primary treatment. Conducted over 2 years, the study included nine female cats diagnosed with Type IV Atresia Ani and recto-vaginal fistula undergoing surgery. Statistical analysis used the R program (version 4.2.1) with Rstudio^®®^ extension. Significant results were observed at a 95% confidence interval and *p* < 0.05. The condition had an incidence of 4.7% among the 192 cats evaluated over a 2-year period. The findings suggest that a body condition score of 3, an age of 3 to 4 weeks, and a fistula length of 1 to 2 mm correlated with better surgical outcomes, reducing the likelihood of fecal incontinence and anal stenosis development, and enhancing defecation awareness during the perioperative period. This study is the first to report the disease incidence in cats undergoing surgery, according to the authors’ knowledge.

## 1. Introduction

Atresia Ani (AA) is a rare congenital anomalie of the rectum and anus development in companion animals. Its incidence has not yet been reported in cats, but in dogs it has been described as 0.007%, with a higher prevalence in purebred breeds, such as Boston Terriers, Poodles, and Cocker Spaniels [1,2]. Females have a 1.79 higher probability compared to males [3]. The embryology of the anorectal region originates from two distinct tissues: endoderm and ectoderm [4]. The hindgut gives rise to the distal third of the transverse, descending, and sigmoid colon; the rectum; and the upper part of the anal canal; with its endoderm also forming the inner lining of the bladder and the urethra. The terminal part of the hindgut opens into the posterior region of the cloaca, the primitive anorectal canal; the allantois opens into the anterior part, the primitive urogenital sinus [5]. The cloaca, present at days 22 to 25 of feline embryo development, is a cavity lined by endoderm and covered on its ventral boundary by superficial ectoderm [6]. The boundary between the endoderm and the ectoderm gives rise to the cloacal membrane [5]. The urorectal septum separates the region between the allantois and the upper intestine. It derives from the fusion of the mesoderm covering the yolk sac and surrounding the allantois. As the embryo grows and the formation of caudal folds continues, the end of the urorectal septum will sit near the cloacal membrane, although the two structures never make contact. The cloacal membrane becomes thinner until it eventually ruptures, thus giving rise to the caudal ending of the intestine [4,5]. There are four types of Anorectal Atresia reported in the literature: (1) Type I: congenital anal stenosis; (2) Type II: imperforated anus; (3) Type III: imperforated anus in combination with the rectum ending in a blind pouch more cranially; and (4) Type IV: discontinuity of the cranial rectum terminating as a blind pouch inside the pelvic canal, with a recto-vaginal fistula in females [2]. Animals that develop rectovaginal fistula defecate through the vulva, potentially presenting vulvar dermatitis, tenesmus, and abdominal distention [7]. However, there is no general agreement on the classifications of atresia, especially regarding Type IV, where communication between the rectum and the urogenital tract may occur [8,9]. Type IV anal atresia occurs predominantly in females due to the differences in embryological development between males and females. During fetal development, the structures of the lower gastrointestinal and urogenital tracts form from a common embryonic structure called the cloaca. The process by which this common structure separates into distinct components for the rectum and urogenital tracts varies between the sexes. In females, the closer anatomical relationship between the developing rectum and the urogenital structures, such as the vagina, can lead to a higher likelihood of developmental anomalies affecting both systems. The rectum and the vaginal canal are situated more closely together and have a more complex interplay during embryogenesis, making the separation process more prone to errors [10,11]. Several diagnostic methods are utilized to identify and evaluate the severity and specifics of the AA condition in dogs and cats. These methods range from physical examination to advanced imaging techniques and are crucial for planning the appropriate surgical intervention [12,13]. The clinical signs of AA typically begin several weeks after birth, especially after weaning, when transitioning from a liquid to a soft/hard diet, or in some cases, in the first few days after birth if there is rejection of the baby by the mother [14]. The main clinical signs to consider are tenesmus, abdominal distension, abdominal discomfort on palpation, bulging/protrusion of the perineum or swelling of the perineum, and stenosis (Type I AA) or an absence of an anal opening (Types II, III, and IV AA). Megacolon may also develop. If there is a concurrent rectovaginal fistula (RvF), feces may pass through the vulva, leading to clinical conditions such as vulvitis, vulvovaginitis, and cystitis [1,15,16,17]. Radiographic imaging with lateral abdominal radiographs is the most common imaging technique routinely used in the diagnosis of this condition. Computed tomography (CT) and magnetic resonance imaging (MRI) are advanced imaging techniques that provides detailed cross-sectional images of the pelvic anatomy, providing crucial information that aids in the comprehensive assessment of this congenital condition. The MRI offers superior soft tissue contrast compared to other imaging techniques. Also, the use of high-resolution ultrasound can also help to identify fistulas, rectal dilatation, and other structural anomalies [18,19,20,21]. Contrast studies are a critical tool in identifying the presence and extent of fistulous connections between the rectum and the genital tract. The process involves the careful administration of a radiopaque contrast agent, which allows for detailed imaging of the fistulous pathways. After the perineal area is cleaned and prepped to maintain a sterile environment, a catheter is gently inserted into the vaginal opening, and the contrast medium is slowly injected. Radiographs are taken immediately after the contrast medium is introduced. In the case of an atresia ani type IV, the contrast medium will be observed flowing from the vagina into the rectal area, highlighting the abnormal connection [22,23,24,25,26]. Surgical intervention is the treatment of choice for resolving any type of Anorectal Atresia. Some authors have also reported the possibility of correcting Type I Anorectal Atresia (anal stenosis) using balloon dilation or bougie dilatation or excision of the stenosed portion of the rectum [2,12,13,16,27,28]. Surgery should be performed as early as possible to avoid potential complications such as the deterioration of body condition, the development of irreversible megacolon, and ascending urinary tract infections [14,29].

This preliminary study was conducted on a sample of female cats and aimed to epidemiologically characterize patients with the clinical condition of Type IV Anorectal Atresia with Rectovaginal Fistula (AAT IV with RvF). It also aimed to evaluate the influence of various parameters, including age, breed, body condition, length of rectovaginal fistula, parity, and regional microbiota, on the perioperative period in these patients.

## 2. Materials and Methods

This retrospective study over a 2-year period utilized nine female cats (*Felis catus*) (N = 9) out of a total of 192 kittens evaluated in primary pediatric consultation during the study period. All the kittens were inpatients at the Anjos of Assis Veterinary Medicine Centre—CMVAA and had been diagnosed with Atresia Ani Type IV (AAT IV) with rectovaginal fistula (RvF) during feline pediatrician consults. All patients underwent blood sample analysis, including a complete blood count, liver and kidney basic biochemistries, and radiographic contrast studies, to visualize fistulous connections between the rectum and the genital tract, prior to surgery to resolve their clinical condition. Their participation was included only after their tutors signed the informed consent form allowing the use of their data. The following parameters were considered for the development of the study database: age; gender; breed; body weight and body condition at the day of surgery, at 8 days post-surgery, and at 30 days post-surgery; parity; presence of anal electrical stimulus reflex; length of the rectovaginal fistula; recovery time; onset of post-surgery defecation; suture removal time; presence of peri-surgical scar anal stenosis; defecation awareness; post-surgical fecal incontinence; and microbiological analysis of samples from the intervened region in the pre- and post-surgery period (T0 = before surgery and T1 = 8 days post-surgery). All female cats evaluated underwent surgical procedures performed by the same surgeon in order to reduce bias. The Fistula Flap Reconstruction technique, adapted from that used in the treatment of dogs, as described by Mahler & Williams, was used. After anesthetic induction, the animal is positioned at the edge of the surgical table in sternal recumbency with the pelvis elevated, and the tail is secured over the lumbar region. Subsequently, the perineal area is clipped and prepared for surgery [27,29]. A urinary catheter 3.5-fr (MILA) is inserted through the vulva and advanced through the rectovaginal fistula to the rectal lumen to serve as a guide. The surgery began with a dorsoventral incision in the soft tissues of the perineal region along the midline, extending from the dorsal commissure of the vulva to the most dorsal zone of the imperforate anus, thus encompassing the caudal part of the fistula and the blind-ended rectum. This allows for good visualization of the area and ample space for perineal reconstruction. Next, the tissues are retracted [27,29]. For the reconstruction of the anal canal and anus, a local flap from the fistula is used without incising it, preserving its communication with the rectum, thus providing normal diameter and length. The fistula, in its dorsal segment to the vaginal orifice, is then horizontally sectioned and debrided from the vagina. The dorsal defect of the vagina is reconstructed through vaginoplasty, using a 5-0 gylconate (Monosyn^®^, B.Braun, Queluz de Baixo, Portugal) interrupted simple sutures. Complete separation between the rectum and vagina is achieved by apposing the soft tissues between the ventral wall of the rectum and the dorsal wall of the vagina. The ventral portion of the anal canal and anus are reconstructed with the fistula flap, while the dorsal and lateral portions are reconstructed using the blind-ended rectal mucosa that had been previously incised. The anal orifice is reconstructed by apposing the rectal mucosa and the fistula flap with the skin using interrupted simple sutures (Figure 1). For surgery, an Aesculight CO_2_ surgical LASER system—VetScalpel^®^ (LightScalpel Inc, Seattle, USA) was used, with the settings of 8W of potency, in a Superpulse mode, and with a 0.25 mm focal point for region dissection. The use of the CO_2_ LASER system limited blood loss during the surgical procedure, facilitating precise dissection of the surrounding tissues. Rectovaginal length was measured three times using a digital caliper (SXG-model 110, Dongguan Hust Tony Instruments Co.^®^, Dongguan, China), and the mean was recorded. As there is no classification system for rectovaginal fistulas in the surgical context, a decision was pre-established to classify them according to their length as from 1 to 2 mm, 2 to 3 mm, or >3 mm, as a criterion to fit the study design and simplify the evaluation of this parameter. All patients received the same pre-surgical medical protocol, considering the use of Methylprednisolone (1 mg/kg IV), Buprenorphine (0.02 mg/kg IM), and Amoxicillin + Clavulanic Acid (10 mg/kg IM). Anesthetic induction was achieved using Ketamine (5 mg/kg IM) and Medetomidine (80 µg/kg IM), and maintenance was achieved using Isoflurane. As outpatient treatment, patients underwent a protocol of Amoxicillin + Clavulanic Acid (10 mg/kg, BID/PO), Tolfenamic Acid (4 mg/kg SID/PO), Reinette Apple Syrup (½ teaspoon BID/PO), Vitamin Complex SID/PO + Topical Vitamin A on sutures BID, Soft diet, and Elizabethan collar until suture removal. There was no need for re-intervention in the anatomical region for any of the patients included on the study. The collected clinical data were compiled into a database in the Microsoft Office 365 Excel program, and statistical analysis was performed using the R program (version 4.2.1) with the Rstudio^®^ extension. The Shapiro–Wilk test was performed to evaluate the normality assumption for most of the studied variables. Inferential statistics relating patient parameters to different post-surgical recovery parameters were performed using Kendall, ANOVA, Fisherman, and Wilcoxon tests. The results were considered statistically significant for a confidence interval of 95% and for all values of *p* < 0.05.

## 3. Results

Table 1 presents the results obtained from descriptive statistics for the parameters assessed in characterizing the sample regarding age, gender, breed, body weight, body condition, fistula size, parity, recovery time, suture removal time, onset of defecation, peri-surgical anal stenosis, anal stenosis dilation, sensation of defecation, and fecal incontinence. Regarding the study sample, this clinical condition had an incidence of 4.7% (9/192 cats). Regarding the parameter of gender, the total sample is represented only by females, with a value of 100%, as only this gender presents Atresia Ani Type IV with rectovaginal fistula (RvF). For age, the total sample had a mean of 4.7 ± 1.3 weeks (minimum of 3 weeks and maximum of 7 weeks), with the most represented age being 5 weeks (33.3%), followed by 3 and 4 weeks (22.2% each one), and then 6 and 7 weeks (each with 11.1%). The Domestic Shorthair breed was the most represented (77.8%), followed by the Persian breed (22.2%). The total sample’s mean body weight was 668 ± 141.97 g (minimum of 453 g and maximum of 952 g). Body condition, graded according to the La Flamme scale (1997), had a mean of 2.78 ± 0.83 (minimum of 2 and maximum of 4) with BC2 being the most represented (44.4%), followed by BC3 (33.3%) and, lastly, BC4 (22.2%). In terms of fistula size, >3 mm was the most represented (55.6%), followed by the size between 2 and 3 mm (33.3%), and, lastly, the size between 1 and 2 mm (11.1%). Regarding parity, individuals without information (ND) accounted for 33.3%. Primiparous represented 50%, and multiparous 50% (2nd, 3rd, and 4th litter, each with 16.6%). Patient recovery time (minutes) averaged 13.2 ± 5.1 min (minimum of 7 min and maximum of 23 min), and suture removal time (days) averaged 7.8 ± 1.2 days (minimum of 6 days and maximum of 10 days). Regarding the parameter of onset of defecation, the total sample was divided into two categories, before 24 h (<24 h) and after 24 h (>24 h), representing 77.8% and 22.2%, respectively. The onset of defecation was observed before 24 h (<24 h) in 77.8% of cases. Peri-surgical anal stenosis and anal stenosis dilation were observed in 66.7% of cases, and no anal stenosis was represented by 33.3% of cases. The sensation of defecation was recorded in 88.9% of cases, while fecal incontinence was observed in 33.3% of cases. Regarding microbiological analysis, cultures were conducted considering two different time points: T0 = pre-surgery and T1 = post-surgery (8 days after). Cultures at T0 showed *Escherichia coli* as the most prevalent bacteria, followed by *Staphylococcus* spp., *Proteus mirabilis*, and *Proteus vulgaris*. At T1, *Escherichia coli*, *Proteus vulgaris*, and *Pseudomonas aeruginosa* were equally represented (Table 1).

In inferential statistics, we chose to investigate the correlation between surgical success parameters including recovery time (in minutes), suture removal time (in days), onset of post-surgery defecation, peri-surgical anal stenosis, anal stenosis dilation, defecation awareness, and fecal incontinence. This examination was based on the following studied parameters: (1) age (in weeks), (2) body condition score (according to the La Flamme scale), (3) fistula length (in millimeters), and (4) parity across the entire sample (Table 2 and Table 3).

Study findings based on body condition score (BCS) were analyzed for recovery time, suture removal time, onset of defecation, peri-surgical anal stenosis, anal stenosis dilation procedure, awareness of defecation, and fecal incontinence (Table 2). The mean recovery times were 12 ± 2.45 min (range: 9–15 min) for BCS 2, 13.67 ± 5.13 min (range: 8–18 min) for BCS 3, and 15 ± 11.31 min (range: 7–23 min) for BCS 4. An ANOVA test indicated no statistically significant differences among these scores (*p*-value = 0.828) (Figure 2). The mean suture removal times were 8 ± 1.41 days (range: 7–10 days) for BCS 2, 8.33 ± 0.58 days (range: 8–9 days) for BCS 3, and 6.5 ± 0.71 days (range: 6–7 days) for BCS 4. The ANOVA test showed no statistically significant differences among the groups (*p*-value = 0.239) (Figure 3). The onset of defecation was observed according to the BCS within 24 h in 100% of cases (*n* = 4) for BCS 2, within 24 h in 66.7% after 24 h in 33.3% of cases for BCS 3, and for both within 24 h and after 24 h in 50% of cases for BCS 4 (n = 1). For the BCS 4, the Fisher’s exact test indicated no statistically significant differences (*p*-value = 0.444) (Figure 4). The BCS 2 and BCS 4 had anal stenosis in 50% (n = 2) and did not in 50% (n = 2). The BCS 3 had anal stenosis in 100% of the individuals (n = 3). The Fisher’s exact test showed no statistically significant differences (*p*-value = 0.5) (Figure 5). Post-surgery awareness of defecation was recorded in 100% of the patients with BCS 2 and BCS 3, and in only 50% with BCS 4. The Fisher’s exact test indicated no statistically significant differences (*p*-value = 0.222) (Figure 6). Fecal incontinence after surgery was observed in 50% of individuals with BCS 2 and 4, and in all individuals (100%) with BCS 3. The Fisher’s exact test showed no statistically significant differences (*p*-value = 0.5) (Figure 7).

Study findings based on age (weeks) were analyzed for recovery time, suture removal time, onset of defecation, peri-surgical anal stenosis, anal stenosis dilation procedure, awareness of defecation, and fecal incontinence (Table 3). The mean recovery times were 15 min for patients of 3 weeks old, 13 ± 7.07 min (range: 8–18 min) for 4-week-old kittens, 11 ± 1.73 min (range: 9–12 min) for 5-week-old kittens, 23 min for 6-week-old kittens, and 7 min for 7-week-old kittens. A Kendall’s test indicated no statistically significant differences among these scores (*p*-value = 0.514) (Figure 8). The mean suture removal times were 7.5 ± 0.71 days (range: 7–8 days) for 3-week-old kittens, 8.5 ± 0.71 days (range: 8–9 days) for 4-week-old kittens, 8.33 ± 1.53 days (range: 7–10 days) for 5-week-old kittens, and 6 and 7 days for 6-week-old and 7-week-old kittens, respectively. The Kendall´s test showed no statistically significant differences among the groups (*p*-value = 0.502) (Figure 9). The onset of defecation was observed within 24 h in 50% (n = 1) and after 24 h in 50% of 3-week-old kittens, and within 24 h in 100% of kittens of 4, 5, 6, and 7 weeks old. The Wilcoxon test indicated no statistically significant differences (*p*-value = 1.0) (Figure 10). Anal stenosis was registered in 50% of the 3-week-old kittens, 66.7% of the 5-week-old kittens, and 100% of the 4-week-old and 6-week-old kittens. The 7-week-old kittens did not present anal stenosis (Figure 10). The Wilcoxon test showed no statistically significant differences (*p*-value = 0.791) (Figure 11). Post-surgery awareness of defecation was recorded in 100% of the kittens of 3, 4, 5, and 6 weeks old. Contrary to this, the 7-week-old kittens did not present awareness of defecation. The Wilcoxon test indicated no statistically significant differences (*p*-value = 0.164) (Figure 12). Fecal incontinence after surgery was observed in 100% of kittens of 3, 4, 6, and 7 weeks old, and in 66.7% of 5-week-old kittens (Figure 13). The Wilcoxon test indicated no statistically significant differences (*p*-value = 0.145) (Figure 12).

Study findings based on fistula length (millimeters) were analyzed for recovery time, suture removal time, onset of defecation, peri-surgical anal stenosis, anal stenosis dilation procedure, awareness of defecation, and fecal incontinence (Table 2). The mean recovery times were 9 min for a fistula length between 1 and 2 mm, 11.67 ± 3.51 min (range: 8–15 min) for a fistula length between > 2 and 3 mm, and 15 6.04 min for a fistula length over 3 mm (range: 7–23 min). An ANOVA test indicated no statistically significant differences among these scores (*p*-value = 0.525) (Figure 14). The mean suture removal times were 8 days (range: 7–8 days) for a fistula length between 1 and 2 mm, 8.67 ± 1.15 days (range: 8–10 days) for a fistula length between > 2 and 3 mm, and 7.2 ± 1.1 days (range: 6–9 days) for a fistula length over 3 mm. The ANOVA test indicated no statistically significant differences among these scores (*p*-value = 0.271) (Figure 15). The onset of defecation was observed within 24 h in 100% (n = 1) for a fistula length between 1 and 2 mm, within 24 h in 66.7% and after 24 h in 33.3% for a fistula length between > 2 and 3 mm, and within 24 h in 80% (n = 4) and after 24 h in 20% for a fistula length of > 3 mm. The Fisher´s test indicated no statistically significant differences (*p*-value = 1.0) (Figure 16). Anal stenosis was registered in 100% for a fistula length between 1 to 2 mm, 66.7% for a fistula length between >2 and 3 mm, and in 60% for a fistula length of > 3 mm. The Fisher´s test showed no statistically significant differences (*p*-value = 1.0). Post-surgery awareness of defecation was recorded in 100% of patients with fistulas of a length between 1 and 2 mm and between >2 and 3 mm, but in 80% for those with fistula lengths of > 3 mm. The Fisher’s test indicated no statistically significant differences (*p*-value = 1.0) (Figure 17). Fecal incontinence after surgery as not registered in any patient with a fistula length between 1 and 2 mm, but it was registered in 33.3% of those with a fistula length between > 2 and 3 mm and 40% of those with a fistula length of > 3 mm. The Fisher´s test indicated no statistically significant differences (*p*-value = 1.0) (Figure 18).

For parity, the total sample was divided into two groups: primiparous (first litter) and multiparous (second, third, and fourth litters). Study findings based on parity were analyzed for recovery time, suture removal time, onset of defecation, peri-surgical anal stenosis, anal stenosis dilation procedure, awareness of defecation, and fecal incontinence (Table 2). The mean recovery times were 12.83 ± 3.87 min (range: 8–18 min) for the primiparous group, and 14 ± 8.19 min (range: 7–23 min) for the multiparous group. An ANOVA test indicated no statistically significant differences among these scores (*p*-value = 0.235) (Figure 19). The mean suture removal times were 8.33 ± 1.03 days (range: 7–10 min) for the primiparous group, and 6.67 ± 0.58 days (range: 6–7 min) for the multiparous group (Figure 20). An ANOVA test indicated no statistically significant differences among these scores (*p*-value = 0.444) (Figure 20). The onset of defecation was observed within 24 h in 83.4% and after 24 h in 16.6% of the primiparous group, and within 24 h in 66.7% and after 24 h in 33.3% of the multiparous group. A Fischer’s test indicated no statistically significant differences among these scores (*p*-value = 0.752) (Figure 21). Anal stenosis was registered in 66.7% of the primiparous and the multiparous groups. The Fisher’s test showed no statistically significant differences (*p*-value = 0.464) (Figure 22). Post-surgery awareness of defecation was recorded in 100% of the primiparous and the multiparous groups. The Fisher’s test indicated no statistically significant differences (*p*-value = 0.333) (Figure 23). Fecal incontinence after surgery was registered in 16.7% of the primiparous group and 66.7% of the multiparous group. The Fisher´s test indicated no statistically significant differences (*p*-value = 0.464) (Figure 24).

## 4. Discussion

Anomalies affecting the rectum and anus in companion animals are relatively uncommon, with Atresia Ani being among those most frequently reported. The low incidence rate, estimated at 0.007% in dogs and unreported in cats [1], coupled with euthanasia at a young age for many affected animals, possibly due to a lack of awareness about the benefits of early surgical intervention [2], contributes to the scarcity of reported cases. The clinical recognition of Atresia Ani is crucial for prompt surgical intervention, which significantly improves patient prognosis. Common clinical signs include tenesmus, abdominal distension, discomfort on abdominal palpation, bulging or protrusion of the perineum, anal stenosis (AA type I), and, in severe cases, even the absence of the anal orifice with feces passing through the vulva (AA type IV) [1,13,15]. Several factors may influence surgical prognosis, such as body condition, age, and fistula length. This preliminary study also investigated the potential correlation between parity, litter affiliation, and surgical prognosis. Tissue healing following primary or secondary intention is influenced by various factors, including tension, pressure, movement, self-mutilation, and, notably, the patient’s overall health status [30]. Factors such as uremia, excessive glucocorticoid administration, hyperadrenocorticism, geriatric status, and malnutrition can delay wound healing [31]. Malnutrition, particularly reduced albumin levels, leads to decreased healing substrates, delaying the healing process. Serum protein levels below 2.0 g/dL have been associated with reduced healing capacity and suture strength, while poor body condition increases anesthetic and surgical risks [31,32]. An observed inverse relationship between body condition score and recovery time was noted in the study. Animals with higher body condition scores did not exhibit shorter recovery times, contrary to expectations, possibly due to drug accumulation in higher body fat animals and the small sample size [21]. The ANOVA test confirmed that there were no statistically significant differences among the different body condition scores. In terms of suture removal time, animals with lower body condition scores (BCS 2 and 3), classified as lean according to the La Flamme scale, experienced delayed tissue healing and postponed suture removal compared to animals with higher body condition scores (BCS 4) [33,34,35]. Anal stenosis is a common postoperative complication, potentially caused by small patient size, edema catheter use, local infection, or suture dehiscence [1]. Body condition influences the anal stenosis occurrence, but no significant differences were found among patients with different body condition scores in the study [36]. Regarding fecal incontinence, individuals with body condition scores of 3 showed a more favorable prognosis compared to those with body condition scores of 2 and 4, aligning with the literature trends, though statistical significance was not observed. Overall, improving patient body condition typically correlates with reduced fecal incontinence risk and enhanced defecation sensation; though not evident in the study, this is possibly due to the young age of some animals.

Age significantly influences surgical prognosis, with early intervention crucial to mitigate complications such as deteriorating body condition, irreversible megacolon, and ascending urinary tract infections. These factors contribute to postoperative complications like urinary and fecal incontinence, anal stenosis, constipation, tenesmus, rectal prolapse, and suture dehiscence [14,27]. Delayed resolution of obstruction may lead to vomiting and dehydration [2], causing esophageal complications, aspiration pneumonia, and renal insufficiency [30,31]. Geriatric patients and those under 1 year experience slower tissue healing due to diminished cutaneous perfusion and an underdeveloped immune system [30,31,37]. Neonates show a reduced drug metabolism due to underdeveloped cytochrome P450 and NADPH systems and an increased body water percentage, impacting the anesthesia response [38]. The recovery time analysis showed an inverse relationship between age and recovery time, albeit a statistically insignificant one [39]. According to the literature findings, hepatic and renal metabolism in neonates is decreased, leading to hepatic accumulation in animals aged ≤ 4 weeks, which prolongs the recovery time. Injectable anesthetic doses may vary depending on age, affecting recovery times [39]. Suture removal times varied according to the age of the patient at the time of the intervention, potentially due to immune system maturation and body condition, with older animals often presenting better body condition [36]. Comparative studies between cats and dogs revealed differences in healing times, possibly due to anatomical factors such as cutaneous angiogenesis and the number of perforating vessels, with cats presenting a slower healing process [32,40,41,42,43,44,45]. While the surgery recovery time and suture removal time may not be directly related to the surgical technique itself, they serve as important indicators of the overall success of the surgical intervention and postoperative management. They are commonly referenced in the surgical literature, clinical guidelines, and quality improvement initiatives as essential components of surgical outcome assessment and monitoring. The study results show that when interventions were performed between the third and fourth weeks of life, no fecal incontinence was recorded, unlike with patients intervened upon at seven weeks of age, where the incidence was only 33%. While no differences were observed in recovery time and fecal incontinence across ages, monitoring these parameters is vital for evaluating surgical success and postoperative care [45,46,47,48,49,50]. Early detection of complications ensures timely intervention and improved patient outcomes. Fecal incontinence can result from congenital anorectal anomalies, depending on the functionality of the external anal sphincter muscle, with surgical iatrogenic injury also being a contributing factor [50,51,52,53].

All individuals intervened upon before the seventh week of age exhibited a sensation of defecation, whereas those intervened upon at the age of seven weeks did not. Despite observations indicating a potential relationship between age and fecal incontinence, statistical analysis did not reveal significant differences. It is important to note that the literature mentions cases where fecal incontinence, albeit initially present, may exhibit a reversible condition, sometimes taking up to approximately 1 year post-surgery to recover [2,53,54]. Similarly, occurrences of anal stenosis varied according to the patient’s age at the time of surgery. All animals intervened upon at the fourth or sixth weeks of age presented stenosis, whereas those intervened upon at seven weeks of age did not present anal stenosis. This could be associated with their larger size compared to the others, aligning with assumptions in the consulted literature [2,50,51,52,53,54,55]. However, no significant difference was found in the statistical analysis. The analysis of fistula length suggested a potential association between larger fistulas and fecal incontinence. This observation suggests a potential association between fistula size and the risk of iatrogenic injury to the innervation in the muscular structure of the external anal sphincter, given that larger fistulas entail a larger intervened area [2,53,54]. However, statistical significance was not observed. Pre-surgical assessment of external anal sphincter functionality was not conducted, which could impact postoperative outcomes [2,50,51,52,53,54,55]. While age and body condition may influence fecal incontinence occurrence, statistical analysis did not support these observations [2,53,54]. Similarly, no significant relationship was found between fistula length and the development of anal stenosis. It was observed that animals with smaller fistulas were more likely to develop anal stenosis postoperatively [2,53,54].

Regarding defecation sensation, only one animal with a fistula larger than 3 mm lacked sensation, but statistical analysis found no significant differences. For post-surgical defecation onset, animals with smaller fistulas (1 to 2 mm) initiated defecation within 24 h post-surgery, while those with larger fistulas (2 to 3 mm and >3 mm) exhibited delayed initiation. This delay may be attributed to reduced local inflammation in animals with smaller fistulas, facilitating quicker recovery and functional return.

When examining parity, the sample was divided into primiparous (first litter) and multiparous (second, third, and fourth litters) groups. While no differences were found in recovery time and post-surgical defecation onset between the groups, variations were noted in defecation sensation, anal stenosis, and fecal incontinence, with the primiparous group generally showing a better prognosis. However, the statistical analysis did not reveal any significant differences between the two groups.

It is notable that the sensation of defecation varied between the primiparous and multiparous groups, with all individuals in the primiparous group experiencing sensation compared to a lower percentage of those in the multiparous group. Similarly, a higher proportion of primiparous individuals did not experience fecal incontinence compared to those in the multiparous group. These differences may be attributed to various factors, such as variations in fistula dimensions (>2 mm) and associated tissue damage. Regarding anal stenosis, a higher proportion of multiparous individuals experienced this condition compared to primiparous individuals. All affected individuals had a fistula dimension exceeding 3 mm, suggesting a possible association between a larger fistula size and an increased risk of anal stenosis. Despite these observations, the statistical analysis did not reveal any significant differences between the two groups.

Microbiological cultures on fecal samples from three animals showed a pre-surgery presence of *E. coli*, *Staphylococcus* spp., and *Proteus mirabilis* in one, *E. coli* and *Staphylococcus* spp. in another, and *E. coli* and *Proteus vulgaris* in the third. Post-surgery, changes in bacterial profiles were observed, with one animal showing only *E. coli*, another showing *Pseudomonas aeruginosa*, and the third showing *Proteus vulgaris*. Regarding post-surgical fecal incontinence, only one animal exhibited it, coinciding with a fistula length greater than 3 mm, a body condition score of 4, and an age of 7 weeks. Bacterial presence included typical commensals like *E. coli* and Staphylococcus, along with *Proteus vulgaris* and *Proteus mirabilis*, known gastrointestinal microbiota members [56,57,58]. Statistical tests did not demonstrate significance, possibly due to the small sample size. In healthy animals, Firmicutes and Bacteroidetes are predominant phyla in the fecal microbiota, with Fusobacteria, Actinobacteria, and Proteobacteria also being prevalent [57,58,59,60,61]. However, statistical tests did not demonstrate statistical significance, possibly due to a type 2 statistical error resulting from the small sample size, indicating insufficient power to assert significance. Retrospective studies provide valuable insights from existing data, ideal for investigating rare diseases or outcomes impractical to study prospectively.

## 5. Conclusions

Atresia Ani, a congenital anomaly affecting the rectum and anus, is rare in cats. Early surgical intervention is crucial to prevent complications such as megacolon and urinary tract infections. Our study found a higher incidence in cats (4.7%) compared to dogs (0.007%). Factors like age, fistula length, and body condition score influence surgical outcomes, although statistical significance was limited by the small sample size, thus being associated with a type 2 statistical error. This study represents the first epidemiological findings of Type IV Atresia Ani in cats undergoing surgery, with ongoing efforts to expand the sample size.

## Figures and Tables

**Figure 1 animals-14-01738-f001:**
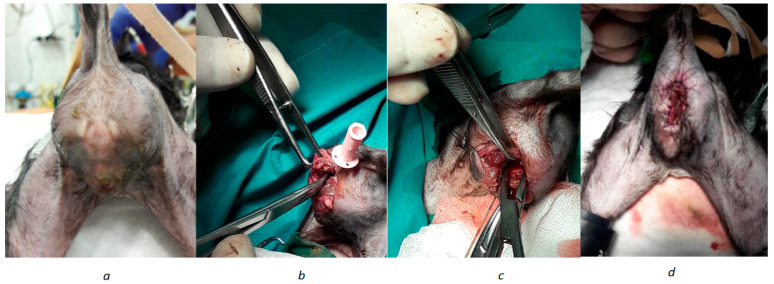
Images of some stages of the surgical procedure performed for the correction of type IV atresia ani. Photograph (**a**) shows the patient′s perineal region and the absence of the anus; (**b**) and (**c**) show dissection work in progress; and (**d**) shows the final appearance of the anoplasty performed on the patient.

**Figure 2 animals-14-01738-f002:**
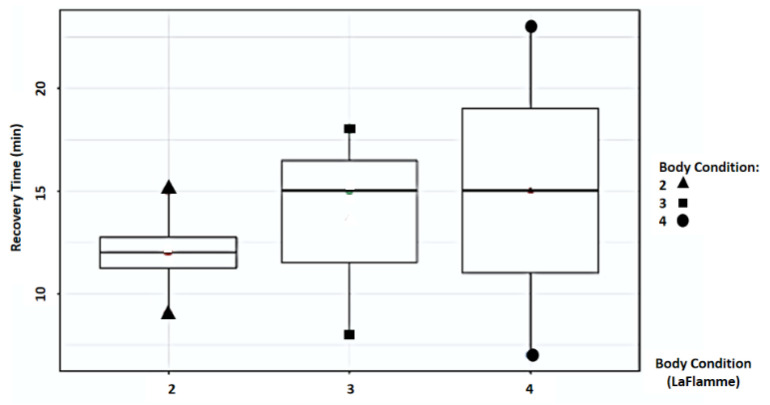
Relationship between body condition score (La Flamme) and recovery time (minutes).

**Figure 3 animals-14-01738-f003:**
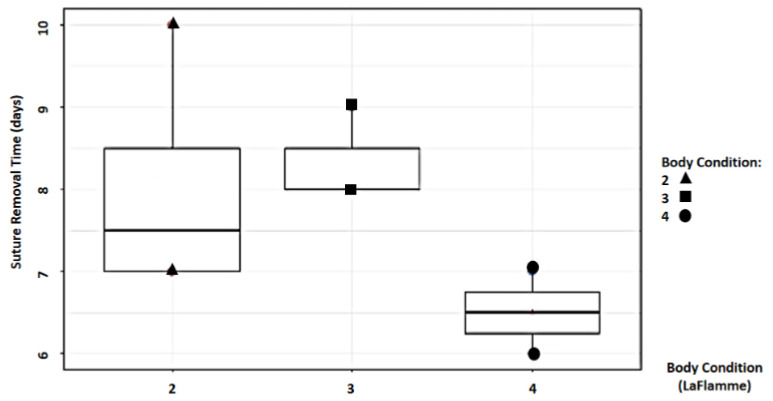
Relationship between body condition score (La Flamme) and suture removal time (minutes).

**Figure 4 animals-14-01738-f004:**
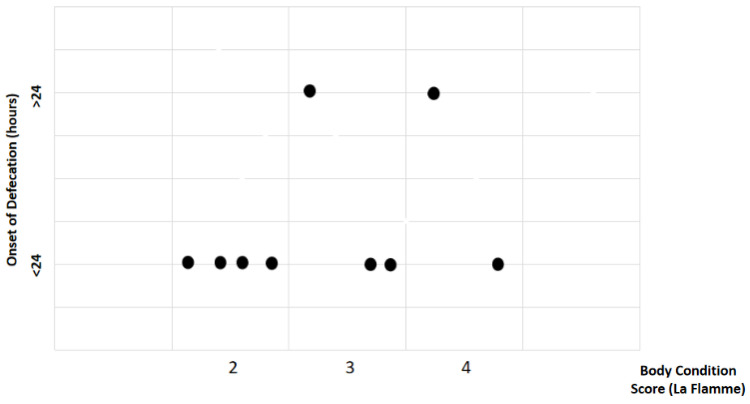
Relationship between body condition score (La Flamme) and onset of defecation. Black dots represent the sample.

**Figure 5 animals-14-01738-f005:**
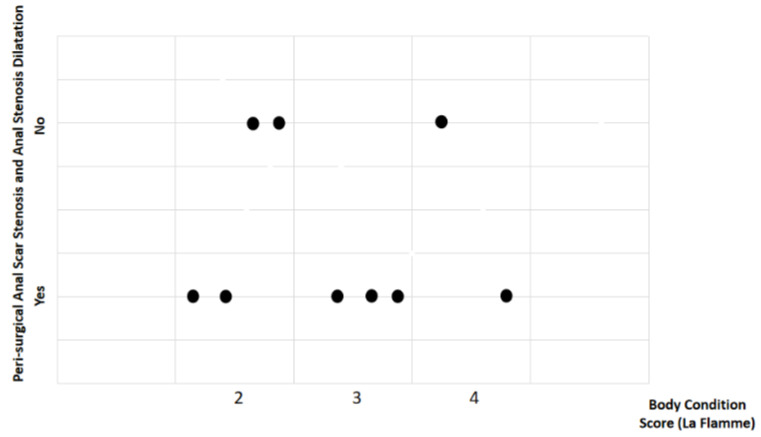
Relationship between body condition score (La Flamme) and peri-surgical anal stenosis and anal stenosis dilation.Black dots represent the sample.

**Figure 6 animals-14-01738-f006:**
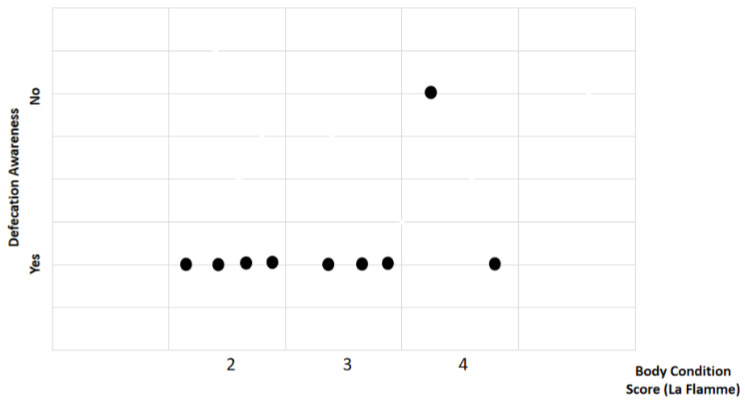
Relationship between body condition score (La Flamme) and awareness of defecation.Black dots represent the sample.

**Figure 7 animals-14-01738-f007:**
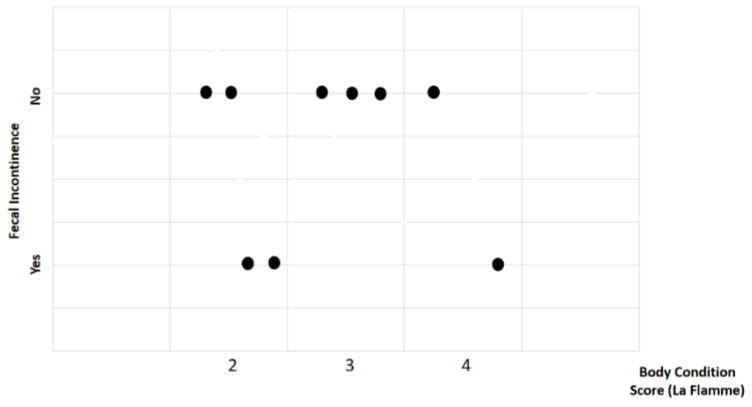
Relationship between body condition score (La Flamme) and fecal incontinence. Black dots represent the sample.

**Figure 8 animals-14-01738-f008:**
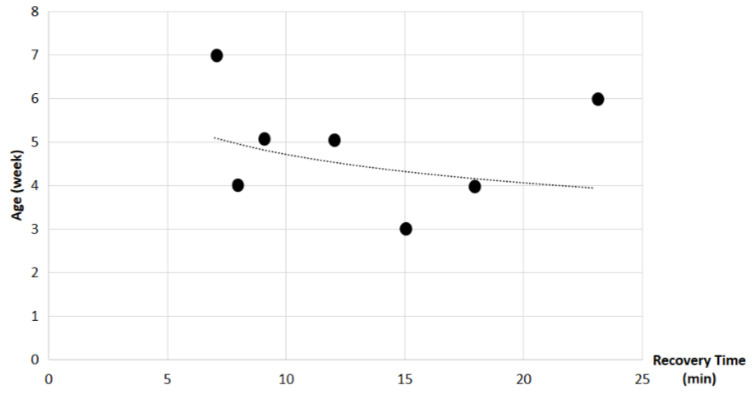
Relationship between age (weeks) and recovery time (minutes). Black dots represent the sample.

**Figure 9 animals-14-01738-f009:**
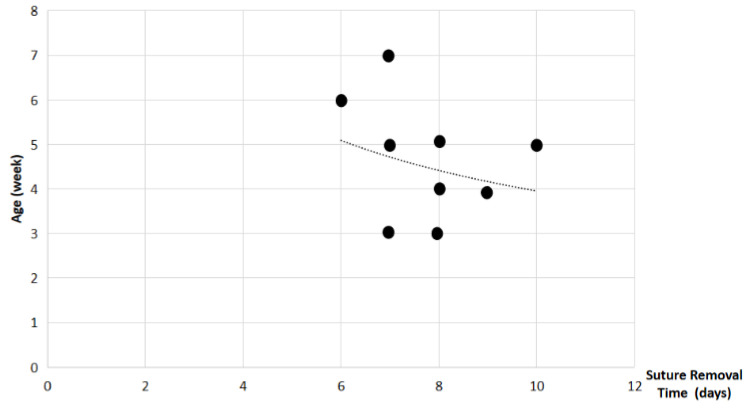
Relationship between age (weeks) and suture removal time (days). Black dots represent the sample.

**Figure 10 animals-14-01738-f010:**
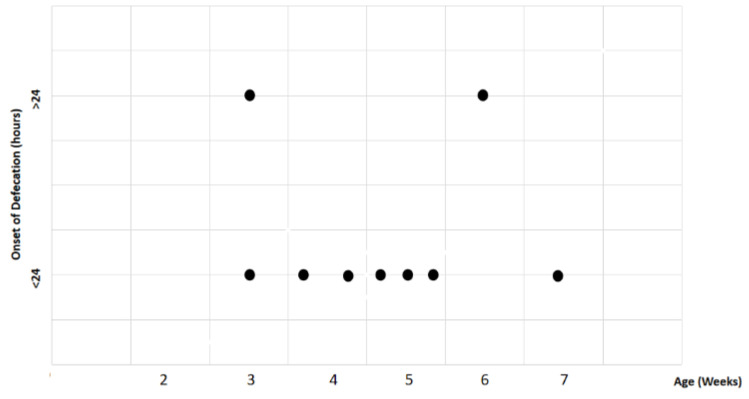
Relationship between age (weeks) and onset of defecation (hours). Black dots represent the sample.

**Figure 11 animals-14-01738-f011:**
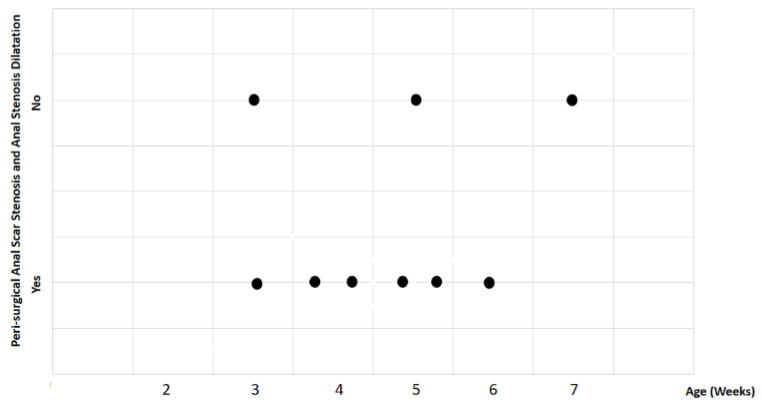
Relationship between age (weeks) and peri-surgical anal stenosis and anal stenosis dilation. Black dots represent the sample.

**Figure 12 animals-14-01738-f012:**
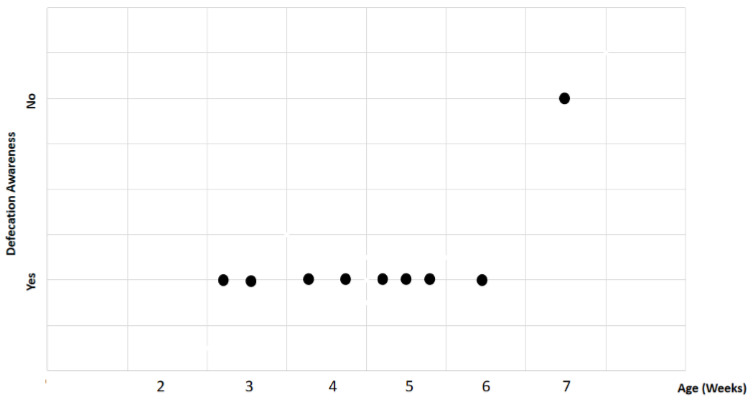
Relationship between age (weeks) and awareness of defecation. Black dots represent the sample.

**Figure 13 animals-14-01738-f013:**
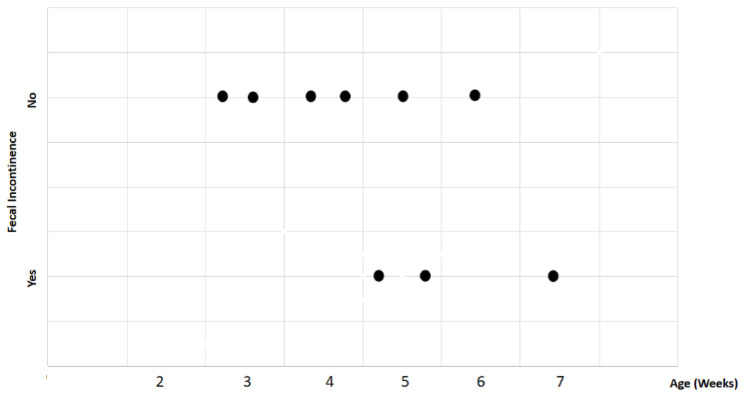
Relationship between age (weeks) and fecal incontinence. Black dots represent the sample.

**Figure 14 animals-14-01738-f014:**
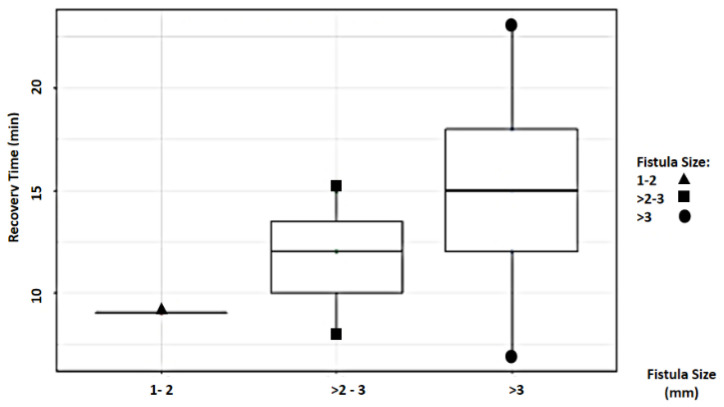
Relationship between fistula length (millimeters) and recovery time (minutes).

**Figure 15 animals-14-01738-f015:**
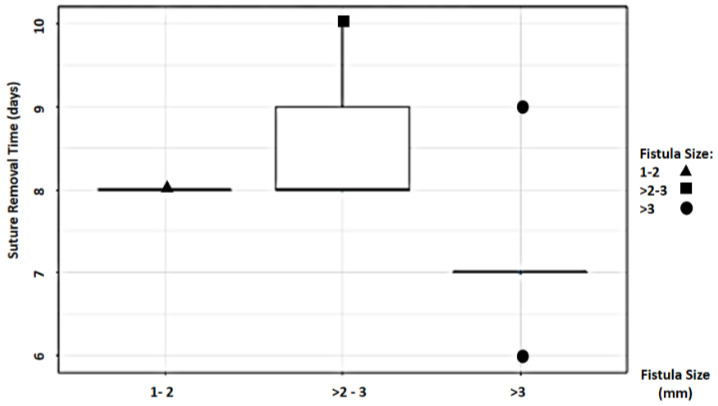
Relationship between fistula length (millimeters) and suture removal time.

**Figure 16 animals-14-01738-f016:**
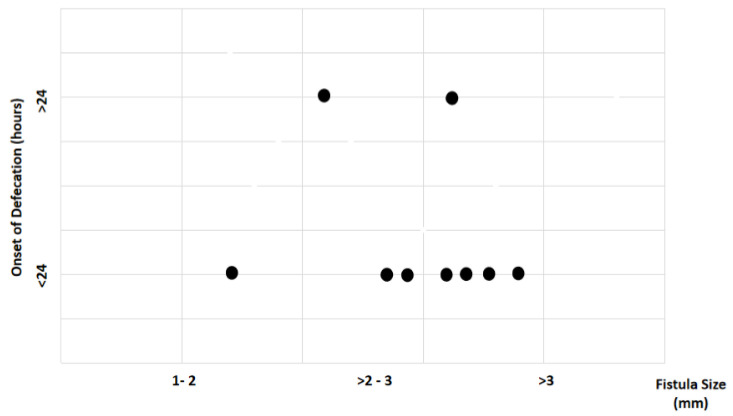
Relationship between fistula length (millimeters) and onset of defecation. Black dots represent the sample.

**Figure 17 animals-14-01738-f017:**
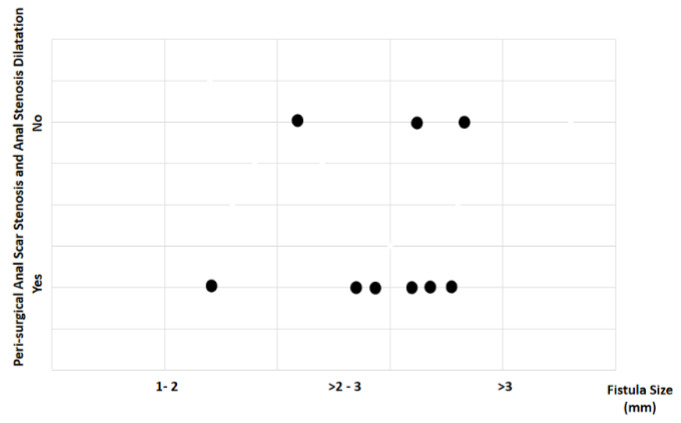
Relationship between fistula length (millimeters) and peri-surgical anal stenosis and anal stenosis dilation. Black dots represent the sample.

**Figure 18 animals-14-01738-f018:**
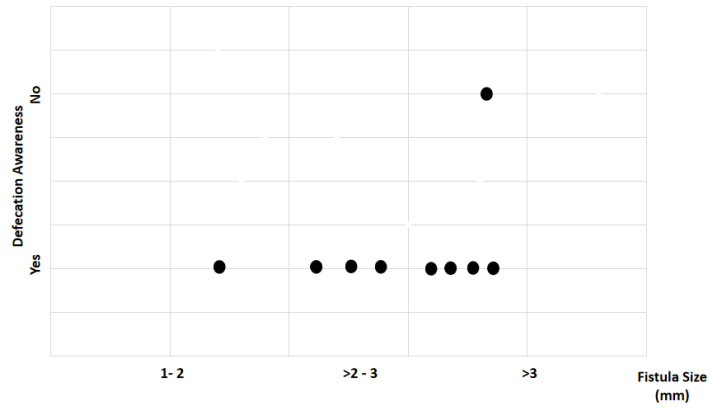
Relationship between fistula length (millimeters) and fecal incontinence. Black dots represent the sample.

**Figure 19 animals-14-01738-f019:**
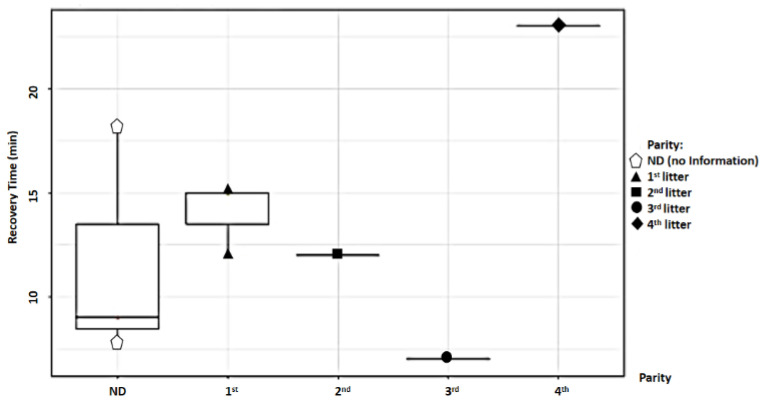
Relationship between parity and recovery time (minutes).

**Figure 20 animals-14-01738-f020:**
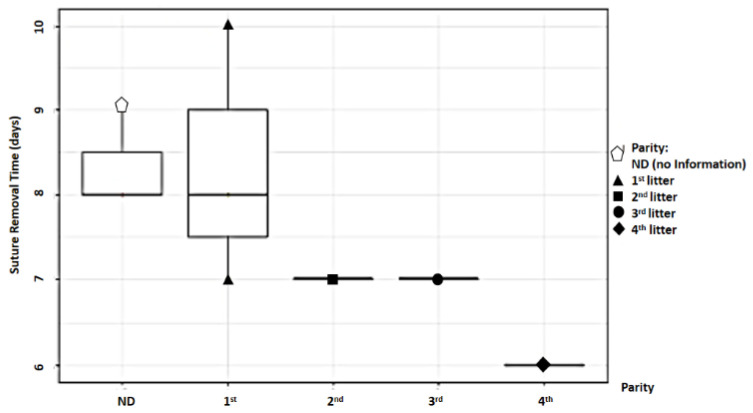
Relationship between parity and suture removal time.

**Figure 21 animals-14-01738-f021:**
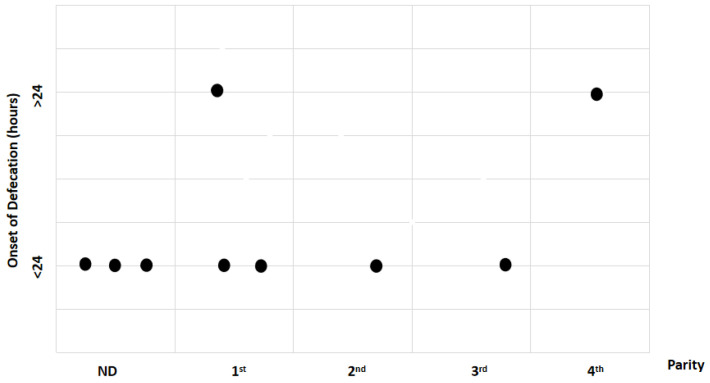
Relationship between parity and onset of defecation. Black dots represent the sample.

**Figure 22 animals-14-01738-f022:**
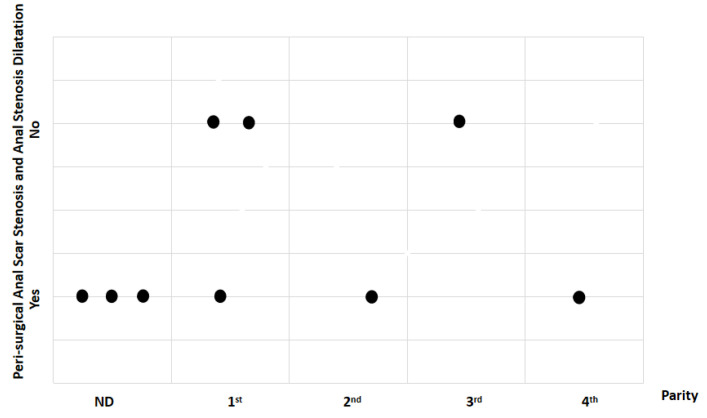
Relationship between parity and peri-surgical anal stenosis and anal stenosis dilation. Black dots represent the sample.

**Figure 23 animals-14-01738-f023:**
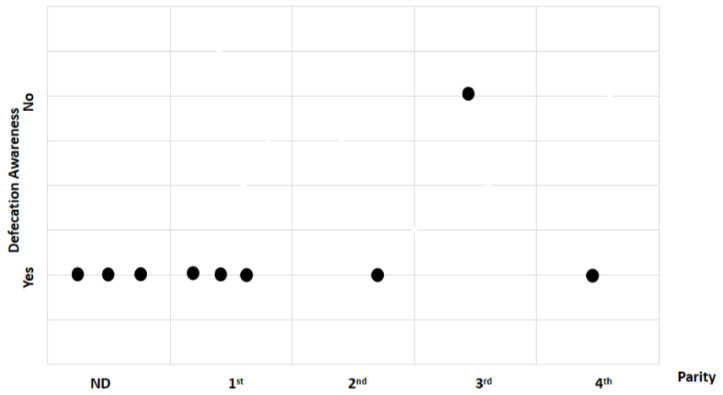
Relationship between parity and awareness of defecation. Black dots represent the sample.

**Figure 24 animals-14-01738-f024:**
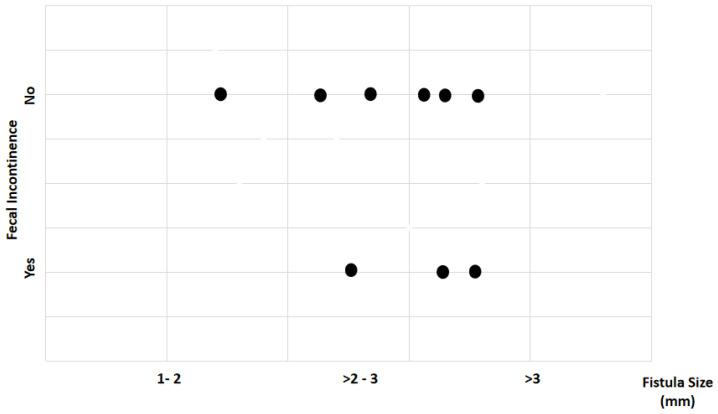
Relationship between parity and fecal incontinence. Black dots represent the sample.

**Table 1 animals-14-01738-t001:** Sample characterization of the several parameters considered on the study.

Parameter		N	$\bar{x}$ ± SD	Shapiro-Wilk (*p*-Value)/Value %
Age (weeks)		9	4.67 ± 1.3	0.544
Gender		9	Female	-
Breed		7	European Common	-
		2	Persian	-
Body Weight (gr)		9	668 ± 141.97	0.804
Body Condition Score (La Flamme)		9	2.78 ± 0.83	0.712
Fistula Length (mm)		1	1–2 mm	11.1%
		3	>2–3 mm	33.3%
		5	>3 mm	55.5%
Parity		3	No Information	33.3%
		3	Primiparous	33.3%
		1	Multiparous (2nd)	11.1%
		1	Multiparous (3rd)	11.1%
		1	Multiparous (4th)	11.1%
Surgery Time (min)		9	69.8 ± 14.7	0.968
Recovery Time (min)		9	13.22 ± 5.14	0.630
Suture Removal Time (days)		9	7.8 ± 1.2	0.588
Onset of Defecation		7	Before 24 h	77.7%
		2	After 24 h	22.2%
Peri-Surgical Anal Scar Stenosis and Anal Stenosis Dilatation		6	Yes	66.6%
		3	No	33.3%
Defecation Awareness		8	Yes	88.8%
		1	No	11.1%
Fecal Incontinence		3	Yes	33.3%
		6	No	66.6%
Microbiological analysis	T0	4	*Escherichia coli*	44.4%
		3	*Staphylococcus* spp.	33.3%
		1	*Proteus mirabilis*	11.1%
		1	*Proteus vulgaris*	11.1%
	T1	3	*Escherichia coli*	33.3%
		3	*Proteus vulgaris*	33.3%
		3	*Pseudomonas aeruginosa*	33.3%

Sample (N); mean ($\bar{x}$); standard deviation (SD); millimeters (mm); gram (gr); minutes (mint); T0 = pre-surgery and T1 = 8 days post-surgery.

**Table 2 animals-14-01738-t002:** Inferential statistical analysis using the ANOVA and Fisherman tests for comparison between body condition score (La Flamme), fistula length, parity, and other parameters.

Parameter		ANOVA Test (*p*-Value)	Fisherman Test (*p*-Value)
Body Condition Score (La Flamme)	Recovery Time (min)	0.828	-
	Suture Removal Time (days)	0.239	-
	Onset of Defecation	-	0.444
	Peri-surgical Anal Scar Stenosis and Anal Stenosis Dilatation	-	0.5
	Defecation Awareness	-	0.222
	Fecal Incontinence	-	0.5
Fistula Length (mm)	Recovery Time (min)	0.525	-
	Suture Removal Time (days)	0.270	-
	Onset of Defecation	-	1
	Peri-surgical Anal Scar Stenosis and Anal Stenosis Dilatation	-	1
	Defecation Awareness	-	1
	Fecal Incontinence	-	1
Parity	Recovery Time (min)	0.235	-
	Suture Removal Time (days)	0.442	-
	Onset of Defecation	-	0.75
	Peri-Surgical Anal Scar Stenosis and Anal Stenosis Dilatation	-	0.463
	Defecation Awareness	-	0.333
	Fecal Incontinence	-	0.464

**Table 3 animals-14-01738-t003:** Inferential statistical analysis using the Kendall and Wilcoxon tests for comparison between age and other parameters.

Parameter		Correlation Value	Kendall Test (*p*-Value)	Wilcoxon Test (*p*-Value)
Age (weeks)	Recovery Time (min)	−0.216	0.514	-
	Suture Removal Time (days)	−0.23	0.502	-
	Onset of Defecation	-	-	1
	Peri-surgical Anal Scar Stenosis and Anal Stenosis Dilatation	-	-	0.791
	Defecation Awareness	-	-	0.164
	Fecal Incontinence	-	-	0.145

## Data Availability

Data are contained within the article.

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
