# Peer review of "Insights into Atresia Ani Type IV in Felis catus: Preliminary Epidemiolocal Findings Associated with Surgery"

_animals, 2024, doi:10.3390/ani14121738_

Round 1

Reviewer 1 Report

Comments and Suggestions for Authors
  • A brief summary (one short paragraph) outlining the aim of the paper, its main contributions and strengths.

This study explores Type IV Atresia Ani in cats, a rare congenital anomaly affecting rectal and anal development, which was previously undocumented. The research focuses on surgical treatment for female cats with this condition, finding that factors like body condition, age, and fistula size influence outcomes. Out of 192 cats evaluated, the incidence of this condition was 4.7%. This study represents the first documentation of Atresia Ani in cats.

  • General concept comments
    Article: highlighting areas of weakness, the testability of the hypothesis, methodological inaccuracies, missing controls, etc.

There are a few identified areas of weakness. One of them is the absence of an accepted complication classification system. The complication outcomes are presented as possibilities.

There are a number of complication classification systems in use:

·         Barrett FM, Bleedorn JA, Hutcheson KD, Torres BT, Fox DB. Comparison of two postoperative complication grading systems after treatment of stifle and shoulder instability in 68 dogs. Vet Surg. 2023 Jan;52(1):98-105. doi: 10.1111/vsu.13893. Epub 2022 Oct 3. PMID: 36189979; PMCID: PMC10092473.;

·         Manekk RS, Gharde P, Gattani R, Lamture Y. Surgical Complications and Its Grading: A Literature Review. Cureus. 2022 May 13;14(5):e24963. doi: 10.7759/cureus.24963. PMID: 35706751; PMCID: PMC9187255.)

In the Summary, the Authors claim: 'This study marks the first documentation of this disease in cats, according to the authors' knowledge.' There is a problem here, as the authors cite a paper (Suess et al., 1992.) which describes a rectovaginal fistula with atresia anal in three kittens. The cited paper may be a case series, and not a study, but the phrasing of the sentence by the authors suggests a different meaning.

  • Is the manuscript clear, relevant for the field and presented in a well-structured manner? 

The manuscript is relevant to the field, although the manuscript is not very clear. The paper is badly written and requires multiple readings to get the message across. Therefore, the presentation of data in this paper needs to be improved.

  • Are the cited references mostly recent publications (within the last 5 years) and relevant? Does it include an excessive number of self-citations?

Most of the references are not recent, but since this is a rare disease with a limited patient pool that cannot be modeled, the selected references are relevant.

  • Is the manuscript scientifically sound and is the experimental design appropriate to test the hypothesis?

Yes. Materials and methods are sound.

The manuscript is scientifically sound, but in this epidemiological study, there is no hypothesis.

It is unclear how and why was the size of the rectovaginal fistula evaluated. Was it divided as: 1 to 2 mm; 2 to 3 mm, and >3 mm on preestablished criteria or for the sake of this study? Is that the common grading system? If yes, where is the reference? Is the said size the diameter or the radius of the fistula?

Was the 8 days post-op the final follow-up time? In such a scarce patient pool, was it not important or beneficial to have longer follow-up times, at least as an e-mail or telephone questionnaire?

  • Are the manuscript’s results reproducible based on the details given in the methods section?

'In inferential statistics, we opted to examine the correlation between surgical success parameters, which include recovery time (in minutes), suture removal time (in days), onset of post-surgery defecation, peri-surgical anal stenosis, anal stenosis dilation, defecation awareness, and fecal incontinence, based on the following studied parameters: 1) Age (in weeks), 2) Body Condition Score (according to La Flamme scale), 3) Fistula size (in millimeters), and 4) Parity, for the entire sample..'

Can the authors explain how surgical recovery time and suture removal time a relevant parameter of surgical success? Is this referenced somewhere? These can be beneficial outcome factors, but are they directly related to the surgery?

What parameters were used to assess the appropriate time of suture removal? This parameter was compared to other parameters, but they are not connected by the presented criteria. Other parameters are observations i.e.- time to defecation and defecation awareness, but suture removal is based on the clinical criteria for suture removal unless pre-determined by the clinician. If predetermined, how can it be measured as an outcome criterion?

  • Are the figures/tables/images/schemes appropriate? Do they properly show the data? Are they easy to interpret and understand? Is the data interpreted appropriately and consistently throughout the manuscript? Please include details regarding the statistical analysis or data acquired from specific databases.

The Figures are appropriate. However, for a paper describing a rare condition, intra-operational photographs would be greatly appreciated. The same can be said for the schematics of the different types of AA and different fistula sizes. This should be addressed by the Authors.

Some presented data is hard to understand, especially paragraph 3.1.

  • Are the conclusions consistent with the evidence and arguments presented?

No.

Discussion, P23L572: 'The type of healing of primary or second intention sutures is affected by various factors, notably tension, pressure, movement, self-mutilation, and the patient's health status, with the latter being the most important.'

Where is this inferred and what impact did this occurence have? This is not mentioned in the results.

Discussion, P23L617:  ..constipation, tenesmus, constipation..

Discussion, P24L641: Studies comparing cats and dogs reported differences in healing 641 times due to cutaneous angiogenesis, whereas in cats, healing at day 7 had half the strength compared to dogs. Reference?

Discussion, P25L685: This situation may be due to the fact that the larger the fistula, the longer the surgical time needed, which in turn is associated with an increase in recovery time.

Was the surgical duration time measured?

Discussion, P27L758: ..consulting in a period of follow-up of 2 years.

This is not clear. In the previous paper text, a follow-up of 8 days is mentioned. I believe the paper encompasses the period of two years, which is not the same as the follow-up period.

  • Please evaluate the ethics statements and data availability statements to ensure they are adequate

Yes. Ethical committee approvals are not necessary for retrospective studies and the Owners reportedly gave informed consent.

Comments on the Quality of English Language

The presentation and readability should be improved.

Author Response

Dear Reviewer,

I hope this message finds you well. Thank you for taking the time to review our manuscript titled "INSIGHTS INTO ATRESIA ANI TYPE IV IN FELIS CATUS: PRELIMINARY EPIDEMIOLOCALLY FINDINGS ASSOCIATED WITH SURGERY," which we have submitted for publication in the Animals. We appreciate your valuable feedback, and we have carefully considered all of your comments. In response, we have made revisions to the manuscript to address each point raised. Below, you will find our detailed responses to your comments, organized to correspond with the changes made in the original version of the manuscript.

Thank you once again for your time and thoughtful input.

Best regards,

General concept comments
Article: highlighting areas of weakness, the testability of the hypothesis, methodological inaccuracies, missing controls, etc. There are a few identified areas of weakness. One of them is the absence of an accepted complication classification system. The complication outcomes are presented as possibilities. There are a number of complication classification systems in use:

Barrett FM, Bleedorn JA, Hutcheson KD, Torres BT, Fox DB. Comparison of two postoperative complication grading systems after treatment of stifle and shoulder instability in 68 dogs. Vet Surg. 2023 Jan;52(1):98-105. doi: 10.1111/vsu.13893. Epub 2022 Oct 3. PMID: 36189979; PMCID: PMC10092473.;

Manekk RS, Gharde P, Gattani R, Lamture Y. Surgical Complications and Its Grading: A Literature Review. Cureus. 2022 May 13;14(5):e24963. doi: 10.7759/cureus.24963. PMID: 35706751; PMCID: PMC9187255.)

Currently, there is no standardized grading system specifically tailored for complications resulting from anal surgery in veterinary medicine, which is why we did not reference one in the manuscript. To address this gap, we presented complication outcomes as possibilities rather than categorizing them using a non-specific system. Although several complication classification systems exist, they are not directly applicable to the context of anal surgery complications in veterinary patients. Therefore, we pre-established a classification framework to fit the needs of our study and to simplify the evaluation of this parameter, ensuring a more accurate and relevant analysis of the complications observed.

In the Summary, the Authors claim: 'This study marks the first documentation of this disease in cats, according to the authors' knowledge.' There is a problem here, as the authors cite a paper (Suess et al., 1992.) which describes a rectovaginal fistula with atresia anal in three kittens. The cited paper may be a case series, and not a study, but the phrasing of the sentence by the authors suggests a different meaning.

This is an omission error from the previous text, as described in the abstract: “This study is the first to report the disease incidence in cats undergoing surgery according to the authors' knowledge”. It has been corrected.

The manuscript is relevant to the field, although the manuscript is not very clear. The paper is badly written and requires multiple readings to get the message across. Therefore, the presentation of data in this paper needs to be improved.

Following the reviewer's suggestions, we attempted to reorganize the manuscript to make it easier and clearer to read

The manuscript is scientifically sound, but in this epidemiological study, there is no hypothesis.

According to the reviewer's suggestions, we replaced the sentence “This preliminary study was developed in a sample of female cats, evaluated and diagnosed with Atresia Ani type IV (AAT IV), and aimed to characterize epidemiologically patients with the clinical condition of AAT IV with a RvF, and to evaluate the influence of multiple parameters such as age, breed, body condition, size of rectovaginal fistula, parity, and regional microbiota on the perioperative period.”, with“This preliminary study was conducted on a sample of female cats and aimed to epidemiologically characterize patients with the clinical condition of Type IV Anorectal Atresia with Rectovaginal Fistula (AAT IV with RvF). It also aimed to evaluate the influence of various parameters, including age, breed, body condition, size of rectovaginal fistula, parity, and regional microbiota, on the perioperative period in these patients.”

It is unclear how and why was the size of the rectovaginal fistula evaluated. Was it divided as: 1 to 2 mm; 2 to 3 mm, and >3 mm on preestablished criteria or for the sake of this study? Is that the common grading system? If yes, where is the reference? Is the said size the diameter or the radius of the fistula?

There is no classification system for fistulas in this context. Therefore, this classification was pre-established as a criteria to fit the study design and simplify the evaluation of this parameter. It should be considered neither the diameter nor the radius, but the length of the fistula, so we added: “...size length of the rectovaginal fistula divided”

Was the 8 days post-op the final follow-up time? In such a scarce patient pool, was it not important or beneficial to have longer follow-up times, at least as an e-mail or telephone questionnaire?

No, the follow-up was not for 8 days post-op. As described in the Material and Methods section the follow-up period for all patients was at least during the first 30 days post-op period: “The following parameters were considered for the development of the study database: age, gender, breed; body weight and body condition at the day of surgery, at 8 days post-surgery, and at 30 days post-surgery.”

Can the authors explain how surgical recovery time and suture removal time a relevant parameter of surgical success? Is this referenced somewhere? These can be beneficial outcome factors, but are they directly related to the surgery?

While these parameters may not be directly related to the surgical technique itself, they serve as important indicators of the overall success of the surgical intervention and postoperative management. They are commonly referenced in surgical literature, clinical guidelines, and quality improvement initiatives as essential components of surgical outcome assessment and monitoring. Thus, monitoring surgical recovery time and suture removal time are considered standard and relevant parameters for evaluating surgical success and postoperative care indicators of healing, for several reasons. The time it takes for a patient to recover from surgery and for sutures to be removed can provide valuable insights into the healing process. Shorter recovery times and earlier suture removal may indicate efficient wound healing and a successful surgical outcome.On the other hand, prolonged recovery times or delayed suture removal can be indicative of complications such as infection, wound dehiscence, or impaired healing. Monitoring these parameters allows for early detection and intervention in case of adverse events.Also, patients with shorter recovery times and earlier suture removal may experience less discomfort and faster return to normal activities, indicating a successful surgical outcome in terms of functional recovery.

What parameters were used to assess the appropriate time of suture removal? This parameter was compared to other parameters, but they are not connected by the presented criteria.

The parameters used to assess the appropriate time of suture removal typically include criteria such as tissue healing, absence of infection, wound stability, and overall clinical judgment. These criteria are often evaluated based on standardized clinical assessments performed by the clinician, which may include visual inspection, palpation, and assessment of wound characteristics.

In the context of comparing suture removal time with other parameters, such as time to defecation and defecation awareness, there may not be a direct connection based on the presented criteria. While suture removal is primarily determined by clinical criteria related to wound healing and stability, time to defecation and defecation awareness may be influenced by other factors such as patient comfort, bowel function, and postoperative care practices. Therefore, while these parameters may be assessed concurrently in a clinical setting, they may not be directly linked by the same criteria for evaluation.

Other parameters are observations i.e.- time to defecation and defecation awareness, but suture removal is based on the clinical criteria for suture removal unless pre-determined by the clinician. If predetermined, how can it be measured as an outcome criterion?

The justification for this statement lies in the nature of the parameters being measured and the clinical practices surrounding suture removal. Time to defecation and defecation awareness are observational parameters that can vary based on individual patient factors and postoperative care. However, suture removal is typically based on standardized clinical criteria, such as tissue healing, absence of infection, and wound stability. If suture removal is predetermined by the clinician, it may still serve as an outcome criterion if it reflects a specific clinical decision-making process. For example, if suture removal is scheduled based on anticipated wound healing milestones or specific postoperative protocols, it can be measured as an outcome criterion to evaluate the success of the predetermined intervention strategy. In this context, the timing of suture removal can provide valuable insights into the effectiveness of the chosen clinical approach and its impact on patient outcomes.

The Figures are appropriate. However, for a paper describing a rare condition, intra-operational photographs would be greatly appreciated. The same can be said for the schematics of the different types of AA and different fistula sizes. This should be addressed by the Authors.

According to the reviewer suggestion we added some intra-operative photographs of the procedure and we for the different types of AA.

Some presented data is hard to understand, especially paragraph 3.1.

As previously mentioned, the manuscript has been revised according to reviewer suggetsion, to make it easier to read and simpler to understand, including a reorganization of the results section.

Discussion, P23L572: 'The type of healing of primary or second intention sutures is affected by various factors, notably tension, pressure, movement, self-mutilation, and the patient's health status, with the latter being the most important.' Where is this inferred and what impact did this occurence have? This is not mentioned in the results.

This justification is inferred from the existing knowledge and principles of wound healing in surgical literature. The impact of these factors on the type of healing (primary or secondary intention) is significant for postoperative outcomes and patient recovery. Although this specific inference may not be explicitly mentioned in the results section, it is commonly understood in the context of surgical practice and management. Therefore, it is essential to acknowledge these factors when discussing surgical techniques, postoperative care, and patient outcomes, even if not explicitly stated in the results section.

Discussion, P23L617:  ..constipation, tenesmus, constipation..

Corrected

Discussion, P24L641: Studies comparing cats and dogs reported differences in healing 641 times due to cutaneous angiogenesis, whereas in cats, healing at day 7 had half the strength compared to dogs. Reference?

The sentence was reformulated and added the references: “Studies comparing cats and dogs reported differences in healing times due to cutaneous angiogenesis and the number of perforating vessels (fewer in cats). In cats, healing at day 7 had half the strength compared to dogs, presenting therefore a slower healing process.”

Discussion, P25L685This situation may be due to the fact that the larger the fistula, the longer the surgical time needed, which in turn is associated with an increase in recovery time. Was the surgical duration time measured?

Yes, data about the surgical procedures duration was registered and so we decided tho add it to table 1.

Discussion, P27L758: ..consulting in a period of follow-up of 2 years.

This is not clear. In the previous paper text, a follow-up of 8 days is mentioned. I believe the paper encompasses the period of two years, which is not the same as the follow-up period.

The reviewer may have overlooked the details provided, so we would like to clarify. During a 2-year data collection period, we evaluated 192 kittens at the feline pediatric clinic. Among them, 9 were diagnosed with type IV anal atresia and underwent surgery. Each patient underwent a minimum follow-up period of 30 days, and not for 8 days as previously explained in question 2.

Reviewer 2 Report

Comments and Suggestions for Authors

To the editor and authors,

The article ‘Insights into Atresia Ani Type IV in Felis catus: Preliminary Epidemiolocally Findings Associated with Surgery’ is an interesting article from a clinical point of view and presents information regarding this pathology in cats. Nevertheless, the surgical technique is not really described and the results are too extensive and could be condensed for a more logical read.

I recommend MAJOR Revisions, once these changes are made, it could be consider for publication.

·         Line 21: should be mentioned why the article describes only females (it is explained later on in the article, but should also appear in the abstract)

·         Line 41-43 is repeating something mentioned in previous lines; please reformulate.

·         Line 67-68 – please mention the difference between males and females.

·         Line 103 – if ‘the Flap Technique’ is adapted from dogs, please mention that (ex: as described by…) and please describe the surgical technique in more detail. Please add pictures to exemplify the surgical procedures performed

·         Line 111 – please reformulate -ex: there was limited blood loss during the surgical procedure

·         Line 119 – were included in the study, not ‘participated’

·         Line 169-173- please reformulate – one can not understand this sentence

·         Line 560 – please compare the study to studies performed in dogs, since in the case of this species there is more information

·         Line 752 – please shorten the conclusions and do not use so much information from the results part

Author Response

Dear Reviewer,

I hope this message finds you well. Thank you for taking the time to review our manuscript titled "INSIGHTS INTO ATRESIA ANI TYPE IV IN FELIS CATUS: PRELIMINARY EPIDEMIOLOCALLY FINDINGS ASSOCIATED WITH SURGERY," which we have submitted for publication in the Animals. We appreciate your valuable feedback, and we have carefully considered all of your comments. In response, we have made revisions to the manuscript to address each point raised. Below, you will find our detailed responses to your comments, organized to correspond with the changes made in the original version of the manuscript.

Thank you once again for your time and thoughtful input.

Best regards,

I recommend MAJOR Revisions, once these changes are made, it could be consider for publication.

Line 21: should be mentioned why the article describes only females (it is explained later on in the article, but should also appear in the abstract)

Corrected: “This study investigated Type IV Atresia Ani (which includes a recto-vaginal fistula) in female cats,...”

Line 41-43 is repeating something mentioned in previous lines; please reformulate.

Indeed, there was a repetition of information. Therefore, it was corrected by the follwoing sentence: “The condition had an incidence of 4.7% among 192 cats evaluated over 2 years period. Findings suggest that a body condition score of 3, age 3 to 4 weeks, and fistula size 1 to 2 millimeters correlated with better surgical outcomes, reducing the likelihood of fecal incontinence and anal stenosis development, and enhancing defecation awareness during the perioperative period.”

Line 67-68 – please mention the difference between males and females.

According to the reviewer's suggestion, we have decided to add the following sentence: “Type IV anal atresia occurs predominantly in females due to the differences in embryological development between males and females. During fetal development, the structures of the lower gastrointestinal and urogenital tracts form from a common embryonic structure called the cloaca. The process by which this common structure separates into distinct components for the rectum and urogenital tracts varies between the sexes. In females, the closer anatomical relationship between the developing rectum and the urogenital structures, such as the vagina, can lead to a higher likelihood of developmental anomalies affecting both systems. The rectum and the vaginal canal are situated more closely together and have a more complex interplay during embryogenesis, making the separation process more prone to errors.”

Line 103 – if ‘the Flap Technique’ is adapted from dogs, please mention that (ex: as described by…) and please describe the surgical technique in more detail. Please add pictures to exemplify the surgical procedures performed

According to the reviewer's suggestion, we decided to describe the surgical technique used in more detail, so we added the following text: “The Fistula Flap Reconstruction technique, adapted from dogs as described by Mahler & Williams, was used. After anesthetic induction, the animal is positioned at the edge of the surgical table in sternal recumbency with the pelvis elevated, and the tail is secured over the lumbar region. Subsequently, the perineal area is clipped and prepared for surgery [27,29]. A urinary catheter 3.5-fr (MILA) is inserted through the vulva and advanced through the rectovaginal fistula to the rectal lumen to serve as a guide. The surgery began with a dorsoventral incision in the soft tissues of the perineal region along the midline, extending from the dorsal commissure of the vulva to the most dorsal zone of the imperforate anus, thus encompassing the caudal part of the fistula and the blind-ended rectum. This allows for good visualization of the area and ample space for perineal reconstruction. Next, the tissues are retracted [27,29]. For the reconstruction of the anal canal and anus, a local flap from the fistula is used without incising it, preserving its communication with the rectum, thus providing normal diameter and length. The fistula, in its dorsal segment to the vaginal orifice, is then horizontally sectioned and debrided from the vagina. The dorsal defect of the vagina is reconstructed through vaginoplasty, using a 5-0 gylconate interrupted simple sutures. Complete separation between the rectum and vagina is achieved by apposing the soft tissues between the ventral wall of the rectum and the dorsal wall of the vagina. The ventral portion of the anal canal and anus are reconstructed with the fistula flap, while the dorsal and lateral portions are reconstructed using the blind-ended rectal mucosa that had been previously incised. The anal orifice is reconstructed by apposing the rectal mucosa and the fistula flap with the skin using interrupted simple sutures. ”

Also, we added pics from surgical procedures (figures 1-4).

Line 111 – please reformulate -ex: there was limited blood loss during the surgical procedure

Following the reviewer's suggestion, we opted to replace the sentence “Surgical work was bloodless, allowing for the dissection of the entire regional tissues in a very accurate way.” by the following “The use of the CO2 LASER system limited blood loss during the surgical procedure, facilitating precise dissection of the surrounding tissues.”

Line 119 – were included in the study, not ‘participated’

Corrected

Line 169-173- please reformulate – one can not understand this sentence

Following the reviewer's suggestion, we reformulated the sentence and replaced it by the following: “ In inferential statistics, we chose to investigate the correlation between surgical success parameters, including recovery time (in minutes), suture removal time (in days), onset of post-surgery defecation, peri-surgical anal stenosis, anal stenosis dilation, defecation awareness, and fecal incontinence. This examination was based on the following studied parameters: 1) Age (in weeks), 2) Body Condition Score (according to La Flamme scale), 3) Fistula size (in millimeters), and 4) Parity, across the entire sample (Tables 2 & 3).”

Line 752 – please shorten the conclusions and do not use so much information from the results part

According to the reviewer's suggestion, we decided to shorten teh Conclusion section: “Atresia Ani, a congenital anomaly affecting the rectum and anus, is rare in cats. Early surgical intervention is crucial to prevent complications such as megacolon and urinary tract infections. Our study found a higher incidence in cats (4.7%) compared to dogs (0.007%), possibly due to  the CMVAA operates as a reference center, allowing for the diagnosis of a higher number of patients with this type of clinical condition. Factors like age, fistula size, and body condition score influence surgical outcomes, although statistical significance was limited by the small sample size, thus being associated with a type 2 statistical error.This study represents the first report of Type IV Atresia Ani incidence in cats undergoing surgery, with ongoing efforts to expand the sample size.”

Reviewer 3 Report

Comments and Suggestions for Authors

The authors are thanked for their work. It is a very interesting work and is well focused and well written.

The only advice that can be given to the authors is that, from the reviewer's point of view, in order to improve their article, given that in the introduction they have made a brief bibliographic review, they should include the different diagnostic methods for type IV anal atresia. The authors indicate that the diagnosis is based on clinical signs and clinical examination, as well as radiographic evaluation. But this is not entirely correct, as contrast radiography and computed tomography are actually the most useful tools in the diagnosis of this pathology.

In addition to indicating, in the material and methods, the diagnostic technique used in the 9 female cats to diagnose anal atresia type IV.

Author Response

Dear Reviewer,

I hope this message finds you well. Thank you for taking the time to review our manuscript titled "INSIGHTS INTO ATRESIA ANI TYPE IV IN FELIS CATUS: PRELIMINARY EPIDEMIOLOCALLY FINDINGS ASSOCIATED WITH SURGERY," which we have submitted for publication in the Animals. We appreciate your valuable feedback, and we have carefully considered all of your comments. In response, we have made revisions to the manuscript to address each point raised. Below, you will find our detailed responses to your comments, organized to correspond with the changes made in the original version of the manuscript.

Thank you once again for your time and thoughtful input.

Best regards,

The only advice that can be given to the authors is that, from the reviewer's point of view, in order to improve their article, given that in the introduction they have made a brief bibliographic review, they should include the different diagnostic methods for type IV anal atresia. The authors indicate that the diagnosis is based on clinical signs and clinical examination, as well as radiographic evaluation. But this is not entirely correct, as contrast radiography and computed tomography are actually the most useful tools in the diagnosis of this pathology.

We completely agree with the reviewer's suggestion that references to the different diagnostic methods should be included. Therefore, we have decided to add the following information in the Introduction section: ”Several diagnostic methods are utilized to identify and evaluate the severity and specifics of the AA condition in dogs and cats. These methods range from physical examination to advanced imaging techniques and are crucial for planning the appropriate surgical intervention. The clinical signs of AA typically begin several weeks after birth, especially after weaning, when transitioning from a liquid to a soft/hard diet, or in some cases, in the first few days after birth if there is rejection of the baby by the mother [12]. The main clinical signs to consider are: tenesmus, abdominal distension, abdominal discomfort on palpation, bulging/protrusion of the perineum or swelling of the perineum, and stenosis (Type I AA) or absence of an anal opening (Type II, III, and IV AA). Megacolon may also develop. If there is a concurrent rectovaginal fistula (RvF), feces may pass through the vulva, leading to clinical conditions such as vulvitis, vulvovaginitis, and cystitis [1,13,14,15]. Radiographic imaging with lateral abdominal radiographs is the most commun imaging technique routinely used in the diagnosis of this condition. Computed Tomography (CT) and magnetic ressonance imaging (MRI) are advanced imaging techniques that provides detailed cross-sectional images of the pelvic anatomy, providing crucial information that aids in the comprehensive assessment of this congenital conditionand. The MRI offers superior soft tissue contrast compared to other imaging techniques. Also, the use of high-resolution ultrasound can also help to identify fistulas, rectal dilatation, and other structural anomalies. Contrast studies are a critical tool in identifying the presence and extent of fistulous connections between the rectum and the genital tract. The process involves the careful administration of a radiopaque contrast agent, which allows for detailed imaging of the fistulous pathways. After the perineal area is cleaned and prepped to maintain a sterile environment, a catheter is gently inserted into the vaginal opening, and the contrast medium is slowly injected. Radiographs are taken immediately after the contrast medium is introduced. In the case of an atresia ani type IV, the contrast medium will be observed flowing from the vagina into the rectal area, highlighting the abnormal connection.”

In addition to indicating, in the material and methods, the diagnostic technique used in the 9 female cats to diagnose anal atresia type IV.

Corrected. We added the following sentence at Material & Methods section: “All patients underwent blood sample analysis, including a complete blood count, liver and kidney basic biochemistries, and radiographic contrast studies to visualize fistulous connections between the rectum and the genital tract, prior to surgery to resolve their clinical condition.”

Round 2

Reviewer 1 Report

Comments and Suggestions for Authors

The Authors addressed some issues regarding the manuscript. There are, however, some issues still unaddressed.

  • General concept comments
    Article: highlighting areas of weakness, the testability of the hypothesis, methodological inaccuracies, missing controls, etc.

There are a few identified areas of weakness. One of them is the absence of the accepted complication classification system. This is still unaddressed.

  • Is the manuscript clear, relevant for the field and presented in a well-structured manner? 

The manuscript has been improved, although there are still some phrasing issues. Consulting a language editor might be beneficial. The Discussion in particular needs further work. Please try to abbreviate the section, with an emphasis on critical points. Discussion written as is exhausts the reader, and is followed by Conclusions which do not get the important message across.

Introduction: Type IV anal atresia occurs predominantly in females due to the differences in embryological development between males and females.

If type IV is a rectovaginal fistula, shouldn't this sentence state 'only' instead of predominantly?

  • Are the cited references mostly recent publications (within the last 5 years) and relevant? Does it include an excessive number of self-citations?

The selected references are relevant.

  • Is the manuscript scientifically sound and is the experimental design appropriate to test the hypothesis?

Yes. Materials and methods are mostly sound.

It is unclear how and why was the size of the rectovaginal fistula evaluated. Was it divided as: 1 to 2 mm; 2 to 3 mm, and >3 mm on preestablished criteria or for the sake of this study? Is that the common grading system? This question has not been answered in the reviewed manuscript. Why is this important for the authors?

P4L132: What brand of glyconate sutures was used?

  • Are the manuscript’s results reproducible based on the details given in the methods section?

'In inferential statistics, we opted to examine the correlation between surgical success parameters, which include recovery time (in minutes), suture removal time (in days), onset of post-surgery defecation, peri-surgical anal stenosis, anal stenosis dilation, defecation awareness, and fecal incontinence, based on the following studied parameters: 1) Age (in weeks), 2) Body Condition Score (according to La Flamme scale), 3) Fistula size (in millimeters), and 4) Parity, for the entire sample..'

Can the authors explain how surgical recovery time and suture removal time a relevant parameter of surgical success? Is this referenced somewhere? These can be beneficial outcome factors, but are they directly related to the surgery? This has not been addressed by the authors.

What parameters were used to assess the appropriate time of suture removal? This parameter was compared to other parameters, but they are not connected by the presented criteria. Others parameters are observations i.e.- time to defecation and defecation awareness, but suture removal is based on the clinical criteria for suture removal unless pre-determined by the clinician. If predetermined, how can it be measured as an outcome criteria? This has not been addressed by the authors.

  • Are the figures/tables/images/schemes appropriate? Do they properly show the data? Are they easy to interpret and understand? Is the data interpreted appropriately and consistently throughout the manuscript? Please include details regarding the statistical analysis or data acquired from specific databases.

The Figures are appropriate. Intraoperative photographs have been provided-however, the description regarding Figures 1-4 is absent. Please look into this.

The previous Figure 1 should now be Figure 5. The previous Figure 2 description doesn't make sense: (Figure 2 - Based on the Body Condition Score (La Flamme) regarding the parameter of Suture removal time (minutes)).

This should be addressed by the Authors.

  • Are the conclusions consistent with the evidence and arguments presented?

No.

Discussion, P20L586: The factors influencing suture healing include the size and location of the suture.

Presumably this reffers to tissue healing. Please check the overall text for such errors.

Discussion, P23L572: 'The type of healing of primary or second intention sutures is affected by various factors, notably tension, pressure, movement, self-mutilation, and the patient's health status, with the latter being the most important.'

Where is this inferred and what impact did this occurence have? This is not mentioned in the results.

Comments on the Quality of English Language

The manuscript has been improved, although there are still some phrasing issues. Consulting a language editor might be beneficial. The Discussion in particular needs further work. Please try to abbreviate the section, with an emphasis on critical points.

Author Response

Dear Reviewer,

I hope this message finds you well. Thank you for taking the time to review our manuscript titled "INSIGHTS INTO ATRESIA ANI TYPE IV IN FELIS CATUS: PRELIMINARY EPIDEMIOLOCALLY FINDINGS ASSOCIATED WITH SURGERY," which we have submitted for publication in the Animals. We appreciate your valuable feedback, and we have carefully considered all of your comments. In response, we have made revisions to the manuscript to address each point raised. Below, you will find our detailed responses to your comments, organized to correspond with the changes made in the original version of the manuscript.

Thank you once again for your time and thoughtful input.

Best regards,

The Authors addressed some issues regarding the manuscript. There are, however, some issues still unaddressed. 

We thank the reviewer for their work, assistance and time in improving the original manuscript, making it easier for readers to understand. Thank you.

There are a few identified areas of weakness. One of them is the absence of the accepted complication classification system. This is still unaddressed.

As we stated previously, currently, there is no standardized grading system specifically tailored for complications resulting from anal surgery in veterinary medicine, which is why we did not reference one in the manuscript. To address this gap, we presented complication outcomes as possibilities rather than categorizing them using a non-specific system. Although several complication classification systems exist, they are not directly applicable to the context of anal surgery complications in veterinary patients. Therefore, we pre-established a classification framework to fit the needs of our study and to simplify the evaluation of this parameter, ensuring a more accurate and relevant analysis of the complications observed.

Is the manuscript clear, relevant for the field and presented in a well-structured manner? The manuscript has been improved, although there are still some phrasing issues. Consulting a language editor might be beneficial. The Discussion in particular needs further work. Please try to abbreviate the section, with an emphasis on critical points. Discussion written as is exhausts the reader, and is followed by Conclusions which do not get the important message across.

The document was reviewed for its grammatical correctness by a native English speaker. No other reviewer mentioned any difficulties in reading or correcting the grammar of the text. The Discussion section was reviewed and rewritten to make it easier to read, according to the reviewer suggestion.

IntroductionType IV anal atresia occurs predominantly in females due to the differences in embryological development between males and females. If type IV is a rectovaginal fistula, shouldn't this sentence state 'only' instead opredominantly?

Considering that hermaphrodites can also have type IV atresia ani, it was decided to use the word predominant. However, if the reviewer prefers, we can use only in females instead.

It is unclear how and why was the size of the rectovaginal fistula evaluated. Was it divided as: 1 to 2 mm; 2 to 3 mm, and >3 mm on preestablished criteria or for the sake of this study? Is that the common grading system? This question has not been answered in the reviewed manuscript. Why is this important for the authors?

We agree with the reviewer that it should be pointed out how the rectovaginal fistulas were measured. Regarding the grading system, the question was already responded in round 1: there is no classification system for fistulas in this context. Therefore, this classification was pre-established as a criteria to fit the study design and simplify the evaluation of this parameter. It should be considered neither the diameter nor the radius, but the length of the fistula, so we added: “...size length of the rectovaginal fistula divided”

We decided to replace the sentence by the following: ”Rectovaginal length was measured three times using a digital caliper (SXG-model 110, Dongguan Hust Tony Instruments Co.®, China), and the mean was recorded. As there is no classification system for rectovaginal fistulas in the surgical context, it was decided to pre-established to classify them according to their length as 1 to 2 mm; 2 to 3 mm, and >3 mm, as a criteria to fit the study design and simplify the evaluation of this parameter.”

What brand of glyconate sutures was used?

We decided to replace the sentence by the following: “...using a 5-0 gylconate (Monosyn®, B.Braun) interrupted simple suture...”

Can the authors explain how surgical recovery time and suture removal time a relevant parameter of surgical success? Is this referenced somewhere? These can be beneficial outcome factors, but are they directly related to the surgery? This has not been addressed by the authors.

Regarding the question how surgical recovery time and suture removal time a relevant parameter of surgical success?, it was already respondedd in round 1:While these parameters may not be directly related to the surgical technique itself, they serve as important indicators of the overall success of the surgical intervention and postoperative management. They are commonly referenced in surgical literature, clinical guidelines, and quality improvement initiatives as essential components of surgical outcome assessment and monitoring. Thus, monitoring surgical recovery time and suture removal time are considered standard and relevant parameters for evaluating surgical success and postoperative care indicators of healing, for several reasons. The time it takes for a patient to recover from surgery and for sutures to be removed can provide valuable insights into the healing process. Shorter recovery times and earlier suture removal may indicate efficient wound healing and a successful surgical outcome.On the other hand, prolonged recovery times or delayed suture removal can be indicative of complications such as infection, wound dehiscence, or impaired healing. Monitoring these parameters allows for early detection and intervention in case of adverse events.Also, patients with shorter recovery times and earlier suture removal may experience less discomfort and faster return to normal activities, indicating a successful surgical outcome in terms of functional recovery.”

What parameters were used to assess the appropriate time of suture removal? This parameter was compared to other parameters, but they are not connected by the presented criteria. Others parameters are observations i.e.- time to defecation and defecation awareness, but suture removal is based on the clinical criteria for suture removal unless pre-determined by the clinician. If predetermined, how can it be measured as an outcome criteria? This has not been addressed by the authors.

This question was already responded in round 1:The parameters used to assess the appropriate time of suture removal typically include criteria such as tissue healing, absence of infection, wound stability, and overall clinical judgment. These criteria are often evaluated based on standardized clinical assessments performed by the clinician, which may include visual inspection, palpation, and assessment of wound characteristics.

In the context of comparing suture removal time with other parameters, such as time to defecation and defecation awareness, there may not be a direct connection based on the presented criteria. While suture removal is primarily determined by clinical criteria related to wound healing and stability, time to defecation and defecation awareness may be influenced by other factors such as patient comfort, bowel function, and postoperative care practices. Therefore, while these parameters may be assessed concurrently in a clinical setting, they may not be directly linked by the same criteria for evaluation.

The Figures are appropriate. Intraoperative photographs have been provided-however, the description regarding Figures 1-4 is absent. Please look into this.

The previous Figure 1 should now be Figure 5. The previous Figure 2 description doesn't make sense: (Figure 2 - Based on the Body Condition Score (La Flamme) regarding the parameter of Suture removal time (minutes)).This should be addressed by the Authors.

We decided to replace the legend of figure 1 from: “Images of some stages of the surgical procedure performed for the correction of type IV atresia ani.” to “Images of some stages of the surgical procedure performed for the correction of type IV atresia ani. Photograph a) shows the patient's perineal region and the absence of the anus; b) and c) dissection work in progress, and d) the final appearance of the anoplasty performed on the patient. “

Also, the numbering of the remaining figures has been corrected, along with their captions.

Discussion, P20L586: The factors influencing suture healing include the size and location of the suture. Presumably this reffers to tissue healing. Please check the overall text for such errors.

Corrected

Discussion, P23L572: 'The type of healing of primary or second intention sutures is affected by various factors, notably tension, pressure, movement, self-mutilation, and the patient's health status, with the latter being the most important.' Where is this inferred and what impact did this occurence have? This is not mentioned in the results.

The sentence appears in the discussion section, aiming to emphasize the importance of providing context for the research and highlighting the relevance of the findings, considering the complex and dynamic process of wound healing that can be influenced by various factors. We decided to replace by “The complex and dynamic process of tissue healing that can following primary or secondary intention is affected by several factors, including tension, pressure, movement, self-mutilation, and most importantly, the patient's overall health status [30].”

Comments on the Quality of English Language. The manuscript has been improved, although there are still some phrasing issues. Consulting a language editor might be beneficial. The Discussion in particular needs further work. Please try to abbreviate the section, with an emphasis on critical points.

The document was reviewed for its grammatical correctness by a native English speaker. No other reviewer mentioned any difficulties in reading or correcting the grammar of the text.

The Discussion section was reviewed and rewritten to make it easier to read, according to the reviewer suggestion.

Reviewer 2 Report

Comments and Suggestions for Authors

Dear authors, thank you for taking my comments into consideration, I believe your article is acceptable for publication. 

Author Response

Dear Reviewer,

I hope this message finds you well. Thank you for taking the time to review our manuscript titled "INSIGHTS INTO ATRESIA ANI TYPE IV IN FELIS CATUS: PRELIMINARY EPIDEMIOLOCALLY FINDINGS ASSOCIATED WITH SURGERY," which we have submitted for publication in the Animals. We appreciate your valuable feedback, and we have carefully considered all of your comments. In response, we have made revisions to the manuscript to address each point raised. Below, you will find our detailed responses to your comments, organized to correspond with the changes made in the original version of the manuscript.

Thank you once again for your time and thoughtful input.

Best regards,

Dear authors, thank you for taking my comments into consideration, I believe your article is acceptable for publication. 

We thank the reviewer for their work, assistance and time in improving the original manuscript, making it easier for readers to understand. Thank you.